# Signatures and Discriminative Abilities of Multi-Omics between States of Cognitive Decline

**DOI:** 10.3390/biomedicines12050941

**Published:** 2024-04-23

**Authors:** Filippos Anagnostakis, Michail Kokkorakis, Keenan A. Walker, Christos Davatzikos

**Affiliations:** 1Department of Medical and Surgical Sciences, Alma Mater University of Bologna, 40126 Bologna, Italy; 2Centre for Biomedical Image Computing and Analytics, University of Pennsylvania, Philadelphia, PA 19104, USA; 3Department of Clinical Pharmacy and Pharmacology, University of Groningen, University Medical Center Groningen, 9700 AB Groningen, The Netherlands; 4Laboratory of Behavioral Neuroscience, National Institute on Aging, Intramural Research Program, Baltimore, MD 21224, USA

**Keywords:** dementia, lipidomics, metabolomics, proteomics, predictions

## Abstract

Dementia poses a substantial global health challenge, warranting an exploration of its intricate pathophysiological mechanisms and potential intervention targets. Leveraging multi-omic technology, this study utilizes data from 2251 participants to construct classification models using lipidomic, gut metabolomic, and cerebrospinal fluid (CSF) proteomic markers to distinguish between the states of cognitive decline, namely, the cognitively unimpaired state, mild cognitive impairment, and dementia. The analysis identifies three CSF proteins (apolipoprotein E, neuronal pentraxin-2, and fatty-acid-binding protein), four lipids (DEDE.18.2, DEDE.20.4, LPC.O.20.1, and LPC.P.18.1), and five serum gut metabolites (Hyodeoxycholic acid, Glycohyodeoxycholic acid, Hippuric acid, Glyceric acid, and Glycodeoxycholic acid) capable of predicting dementia prevalence from cognitively unimpaired participants, achieving Area Under the Curve (AUC) values of 0.879 (95% CI: 0.802–0.956), 0.766 (95% CI: 0.700–0.835), and 0.717 (95% CI: 0.657–0.777), respectively. Furthermore, exclusively three CSF proteins exhibit the potential to predict mild cognitive impairment prevalence from cognitively unimpaired subjects, with an AUC of 0.760 (95% CI: 0.691–0.828). In conclusion, we present novel combinations of lipids, gut metabolites, and CSF proteins that showed discriminative abilities between the states of cognitive decline and underscore the potential of these molecules in elucidating the mechanisms of cognitive decline.

## 1. Introduction

Dementia is a growing health concern and presents a rising burden of morbidity and mortality, with an estimated prevalence increase from 57.4 million in 2019 to 152.8 million in 2050 [1]. Beyond established genetic predispositions, the anticipated increase in dementia cases underscores the importance of addressing cardiometabolic risk and the need for primary and secondary prevention [2].

Intense research has focused on elucidating molecular differences between cognitively healthy and demented individuals. Large-scale multi-omic studies reveal pivotal insights into Alzheimer’s Disease (AD), offering the potential for personalized diagnosis and treatment. Despite their contributions, challenges such as AD complexity and cohort heterogeneity need to be addressed to advance our understanding. The integration of data from various omics platforms provides valuable insights into intricate biological pathways, paving the way for targeted therapeutic interventions and precision medicine [3].

The dysregulation of plasma lipidome is evident in AD and offers great predicting capabilities for prevalent and incident cases of AD. A wide range of lipids is suggested to be implicated in the disease’s pathophysiology, such as phospholipids, sulfatides, gangliosides, ceramides, and plasmalogens [4,5].

The proteomic signatures associated with AD subtypes were present already at the preclinical stage and largely remained stable with increasing disease severity [6]. It has been shown that SMOC1 and SPON1 proteins are associated with Aβ deposition and brain structural integrity, which are present in high concentrations nearly 30 years before the onset of symptoms. The study of the temporal evolution of AD alongside proteomic changes may seem useful in the spectrum of early detection of AD pathophysiological changes and subsequently reveal novel diagnostic and therapeutic perspectives [7].

Studies have revealed a significant link between gut metabolomic changes and Alzheimer’s disease (AD). Key findings include altered gut microbial metabolites, with differences in tryptophan metabolites, short-chain fatty acids (SCFAs), and lithocholic acid observed in AD patients compared to controls. Gut microbiome alterations in AD participants showcase decreased microbial diversity and a distinct composition, emphasizing the gut–brain axis’s potential influence on the central nervous system function [8]. A mouse model study associates dysregulated the gut–brain axis involvement with AD progression, highlighting the need to characterize dysbiosis for alternative disease management strategies [9]. Collectively, these studies underscore the intricate interplay between gut metabolites, blood lipids, and CSF proteomics, with AD pathophysiology, thereby offering insights into potential biomarkers, therapeutic targets, and lifestyle interventions for AD management. Understanding these processes at early stages is crucial for early diagnosis, prognosis, and designing targeted interventions.

Considering the profound impact of molecular profiles on cognitive decline, this study investigates alterations and diagnostic abilities of baseline blood lipidomic, serum gut microbial metabolomic, and cerebrospinal fluid (CSF) proteomic profiles in cognitively normal individuals for identifying the prevalence of dementia.

## 2. Materials and Methods

In this study of the Alzheimer’s Disease Neuroimaging Initiative (ADNI), 2251 participants with a baseline diagnosis of cognitively unimpaired (CU), mild cognitive impairment (MCI), and dementia were included, of which the available lipidomic, metabolomic, and CSF proteomic features were obtained. At baseline, it included 677 CU individuals, 1203 with a diagnosis of MCI, and 449 with dementia. After filtering for missing values, the sample numbers used for the statistical analyses are shown in Table 1.

The ADNI was launched in 2003 as a public–private partnership, led by Principal Investigator Michael W. Weiner, MD. The primary goal of ADNI has been to test whether serial MRI, PET, other biological markers, and clinical and neuropsychological assessments can be combined to measure the progression of mild cognitive impairment (MCI) and early Alzheimer’s disease. The ADNI participants were recruited from >50 sites across the United States and Canada. Detailed descriptions of the diagnostic criteria for ADNI have been reported in previous publications [10]. Study data were obtained from the ADNI database (https://adni.loni.usc.edu/, assessed on 21 December 2023).

### 2.1. Lipidomics Data

A lipidomic analysis targeting 781 specific lipid species was conducted using plasma samples from participants in the ADNI study. In total, 10 μL of pre-portioned plasma was combined with 100 μL of butanol–methanol (1:1) solution containing 10 mM of ammonium formate and a blend of internal standards. Following vortexing and sonication, the samples were centrifuged at 14,000× *g* for 10 min at 20 degrees Celsius before being transferred into sample vials with glass inserts for analysis. This involved utilizing ultra-high-performance liquid chromatography combined with chromatographic separation techniques to distinguish between isomeric and isobaric lipid species. Mass spectrometry analysis was carried out using an Agilent mass spectrometer (model 6490 QQQ) operating in positive ion mode, employing dynamic scheduled multiple reaction monitoring (MRM). The analysis protocol followed was developed by Kevin Huynh and Peter Meikle at the Metabolomics Laboratory of the Baker Heart and Diabetes Institute. Detailed information about their lipidomics platform can be found in the methodology file (https://adni.bitbucket.io/reference/docs/ADMCLIPIDOMICSMEIKLELABLONG/meiklelab_methods_[Updated_for_20-10-2020].pdf, assessed on 21 December 2023) and related publications [5].

### 2.2. Gut Metabolomics

The gut microbial metabolomic analysis was conducted by an ultra-performance liquid chromatography coupled to tandem mass spectrometry (UPLC-MS/MS) system (ACQUITY UPLC-Xevo TQ-S, Waters Corp., Milford, MA, USA), and 104 metabolites were quantified in human serum samples. For sample preparation, 20 μL of each standard solution or serum sample was combined with 120 μL of an internal standard solution. Following centrifugation at 13,500× *g* and 4 degrees Celsius for 10 min, 30 μL of the supernatant was transported to a 96-well plate for derivatization. Each well received a 10 μL aliquot of freshly prepared derivative reagents: 200 mM 3-NPH in 75% aqueous methanol and 96 mM EDC-6% pyridine solution in methanol. After derivatization at 30 degrees Celsius for 60 min, 400 μL of ice-cold 50% methanol solution was added to dilute the sample. The plate was then stored at −20 degrees Celsius for 20 min, followed by centrifugation at 4 degrees Celsius for 30 min at 4000× *g*. Subsequently, 135 μL of supernatant was transferred to a new 96-well plate in each well. Lastly, the plate was sealed for LC-MS analysis, with an injection volume of 5 μL. The analysis was conducted following the protocol developed by Huizhen Zhang at the University of Hawaii Cancer Center. A detailed description of their gut microbial metabolomic platform was provided in the methodology file (https://adni.bitbucket.io/reference/docs/ADMCGUTMETABOLITESLONG/Uhawaii_methods_Human_serum_Rima_MicrobiomeMetabolites[Updated_for_20-10-2020].pdf, assessed on 21 December 2023) and respective articles [11].

### 2.3. CSF Proteomics

Targeted CSF proteomics data were analyzed and processed by the Department of Neurology, Emory University School of Medicine, using CSF samples from the ADNI cohort obtained by mass spectrometry. This study encompassed 306 cerebrospinal fluid (CSF) samples. Each CSF sample, amounting to 0.5 mL, was preserved at −80 °C. Upon thawing, 100 µL of each sample underwent depletion of high-abundance proteins utilizing a MARS14 immunoaffinity resin (4.6 × 100 mm column, Agilent, Santa Clara, CA, USA), and the lower abundance proteins were then stored at −80 °C. Furthermore, the frozen samples underwent lyophilization over a period of 72 h and were subjected to overnight digestion with trypsin at a protease-to-protein ratio of circa 1:25. Following digestion, the samples were once again lyophilized and desalted using a 3M Empore C18 96-well plate. Two sets of replicate mass spectrometry (MS) plates were prepared for each sample, which were then dried by vacuum evaporation and stored at −20 °C before MS analysis. The samples were reconstituted with a solution containing 5 internal standard peptides, followed by LC/MRM-MS analysis performed on a QTRAP 5500 instrument. Detailed information regarding protein assessment and quality control can be found at the provided link. In the case of ADNI MRM data, the finalized ‘Normalized Intensity’ data, which underwent quality control procedures, were utilized, with further explanation of the normalization process available in the “Biomarkers Consortium CSF Proteomics MRM data set” within the “Data Primer” document at adni.loni.ucla.edu and in related publications [12].

### 2.4. APOE Genotyping

At the baseline visit, blood samples were obtained from the participants, shipped to the central biomarker analysis lab at the University of Pennsylvania, and processed using an APOE genotyping kit, as further described (http://adni.loni.usc.edu/wp-content/uploads/2010/09/ADNI_GeneralProceduresManual.pdf, assessed on 21 December 2023). The APOE genotype was evaluated by examining two SNPs (rs429358, rs7412) that characterize the epsilon 2, 3, and 4 alleles. This analysis was conducted using DNA extracted from a 3 mL portion of EDTA blood.

### 2.5. Statistical Analysis

In the multi-omic analyses, log10 transformation followed by a standard normalization (zero mean and one-unit standard deviation) was performed on each multi-omic variable. Lipidomic features, owing to their large variable number, were filtered by the lower 25% of Relative Standard Deviation (RSD). Feature selection was performed using ten-fold cross-validated Recursive Feature Elimination (RFE). The top 5 contributing variables were extracted. Then, we included only the variables adjusted for multicollinearity with a variance inflation factor (VIF) < 3 and performed logistic regression with stratified ten-fold cross-validation to assess the models’ discriminative performance in terms of the area under the receiver operating characteristic curve (AUC). Furthermore, we used Youden’s Index to determine the optimal risk threshold. All analyses were performed using R version 4.3.2 (31 October 2023).

## 3. Results

The baseline characteristics of the study population are presented in Table 1. After RFE, four lipids (DEDE.18.2, DEDE.20.4, LPC.O.20.1, and LPC.P.18.1), five serum gut metabolites (Hyodeoxycholic acid, Glycohyodeoxycholic acid, Hippuric acid, Glyceric acid, and Glycodeoxycholic acid), and three CSF proteins (apolipoprotein E, neuronal pentraxin-2, and fatty-acid-binding protein) were included.

Compared to blood lipids and metabolites, CSF proteins differentiated CU individuals from those with dementia with the highest AUC (0.879, 95% CI: 0.802–0.956) (Table 2, Table 3, Table 4 and Table 5. Blood lipids showed an AUC of (0.766, 95% CI: 0.700–0.835), and gut metabolites had an AUC of (0.717, 95% CI: 0.657–0.777). All three categories (blood lipids, CSF proteins, and blood metabolites) demonstrated similar performances in differentiating between MCI and dementia, with AUCs ranging between 0.617 and 0.673 (Table 2, Table 6, Table 7 and Table 8). Ultimately, to distinguish individuals with CU from those who developed MCI, CSF proteins remained first in terms of performance, with an AUC of 0.760 (95% CI: 0.691–0.828), followed by blood lipids with an AUC of 0.655 (95% CI: 0.610–0.701) and gut metabolites with an AUC of 0.556 (95% CI: 0.516–0.595) (Table 2, Table 9, Table 10 and Table 11). The inclusion of the number of APOE4 alleles did not significantly increase the AUC in any prediction.

## 4. Discussion

In our investigation, the discriminative abilities of biomarker panels derived from CSF, lipidomic, and metabolomic profiles for the prognosis of prevalent dementia were evaluated. The inclusion of three CSF proteins (apolipoprotein E, neuronal pentraxin-2, and fatty-acid-binding protein) demonstrated excellent discriminative capabilities between CU individuals and individuals with dementia, while four lipids and five metabolites exhibited significant, albeit less pronounced, discriminative abilities. Notably, none of the feature groups studied in this study met clinical significance in differentiating between MCI and dementia and between CU and MCI.

It is known that there are differences in circulating plasma and CSF molecules between cognitively unimpaired individuals and those with cognitive decline. However, the available literature is still limited and presents heterogeneity [13,14,15]. Lysophosphatidylcholines and dehydrodesmosterol ester alterations are documented in AD pathology and are associated with its progression [16]. Regarding CSF proteomics, apolipoprotein E, neural pentraxin-2, and fatty-acid-binding protein have been implicated in neurodegeneration, which are markers of prognosis [7], synaptic function, amyloid deposition [17], and neuronal membrane disruption [18], respectively. The relationship between the utilized gut metabolites and neurodegeneration is yet to be elucidated. There is early evidence that multi-omic features, such as beta amyloid and tau, may be associated with the CSF biomarkers of neurodegeneration [4]. In light of our findings, further research is necessary to comprehensively study the pathophysiological pathways of cognitive decline using proteomic, lipidomic, and metabolomic markers.

Acknowledging the absence of an external validation cohort as a noteworthy constraint warranting consideration in future investigations is essential. In conclusion, we present novel combinations of lipids, metabolites, and proteins that showed discriminative abilities between the states of cognitive decline and underscore the potential of these molecules in elucidating the mechanisms of cognitive decline.

## Figures and Tables

**Table 1 biomedicines-12-00941-t001:** Baseline characteristics.

	Lipidomics	Gut Metabolomics	CSF Proteomics
Participants (n)	883	1168	278
Age (SD)	73 (±7)	74 (±7)	75 (±7)
Sex			
Female (n)	491	678	172
Male (n)	391	490	106
Diagnosis			
Cognitively Unimpaired (n)	245	352	80
Mild Cognitive Impairment (n)	482	598	123
Dementia (n)	156	218	75
Number of ApoE4 Alleles			
None (n)	475	623	142
One (n)	321	433	99
Two (n)	87	112	37
Education years (SD)	16 (±3)	16 (±3)	16 (±3)
BMI category			
Normal weight (n)	288	421	121
Overweight (n)	402	534	120
Obese (n)	193	213	37

Abbreviation: SD, standard deviation; CSF, cerebrospinal fluid; ApoE4, apolipoprotein E4; BMI, body mass index.

**Table 2 biomedicines-12-00941-t002:** Comparison of AUC (95% CI) performance for each set of predictors.

Predictors	CU vs. Dementia	MCI vs. Dementia	CU vs. MCI
Blood Lipids − DEDE.18.2 − DEDE.20.4 − LPC.O.20.1 − LPC.P.18.1	0.766 (0.700–0.835)	0.617 (0.581–0.652)	0.655 (0.610–0.701)
CSF proteins − Apolipoprotein E − Neuronal pentraxin-2 − Fatty-acid-binding protein	0.879 (0.802–0.956)	0.673 (0.594–0.752)	0.760 (0.691–0.828)
Blood gut Metabolites − Hyodeoxycholic acid − Glycohyodeoxycholic acid − Hippuric acid − Glyceric acid − Glycodeoxycholic acid	0.717 (0.657–0.777)	0.653 (0.611–0.695)	0.556 (0.516–0.595)

Abbreviations: cognitively unimpaired, CU; mild cognitive impairment, MCI; CSF, cerebrospinal fluid.

**Table 3 biomedicines-12-00941-t003:** Prediction of cognitive unimpaired—Dementia with lipids.

Predictors	AUC (95%CI)	Standard Deviation	Accuracy	Precision	Recall	F1	Youden Index	SE	NPV	PVV
4 lipids	0.766 (0.700–0.835)	0.097	0.650	0.545	0.400	0.461	0.270 (0.160–0.380)	0.049	0.690	0.550
4 lipids + APOE4 alleles	0.836 (0.798 to 0.874)	0.053	0.718	0.667	0.533	0.593	0.227 (0.119–0.336)	0.048	0.741	0.667

**Table 4 biomedicines-12-00941-t004:** Prediction of cognitive unimpaired—Dementia with metabolites.

Predictors	AUC (95%CI)	Standard Deviation	Accuracy	Precision	Recall	F1	Youden Index	SE	NPV	PPV
5 metabolites	0.717 (0.657–0.777)	0.083	0.632	0.533	0.364	0.432	0.127 (0.016–0.239)	0.050	0.667	0.533
5 metabolites + APOE4 alleles	0.794 (0.751–0.837)	0.060	0.825	0.833	0.682	0.750	0.179 (0.078–0.280)	0.044	0.821	0.833

**Table 5 biomedicines-12-00941-t005:** Prediction of cognitive unimpaired—Dementia with CSF proteins.

Predictors	AUC (95%CI)	Standard Deviation	Accuracy	Precision	Recall	F1	Youden Index	SE	NPV	PPV
3 CSF proteins	0.879 (0.802–0.956)	0.107	0.875	0.875	0.875	0.875	0.660 (0.503–0.818)	0.070	0.875	0.875
3 CSF proteins + APOE4 alleles	0.891 (0.836–0.946)	0.076	0.933	1	0.857	0.923	0.614 (0.428–0.800)	0.082	0.889	1

**Table 6 biomedicines-12-00941-t006:** Prediction of mild cognitive impairment—Dementia with lipids.

Predictors	AUC (95%CI)	Standard Deviation	Accuracy	Precision	Recall	F1	Youden Index	SE	NPV	PVV
4 lipids	0.623 (0.533–0.712)	0.125	0.750	0.500	0.063	0.111	0.045 (0.007–0.082)	0.017	0.759	0.500
4 lipids + APOE4 alleles	0.661 (0.603–0.719)	0.081	0.770	0.667	0.125	0.211	0.065 (−0.021–0.109)	0.020	0.774	0.667

**Table 7 biomedicines-12-00941-t007:** Prediction mild cognitive impairment—Dementia with metabolites.

Predictors	AUC (95%CI)	Standard Deviation	Accuracy	Precision	Recall	F1	Youden Index	SE	NPV	PPV
5 metabolites	0.653 (0.611–0.695)	0.059	0.732	0.500	0.045	0.083	0.037 (0.003–0.070)	0.015	0.738	0.500
5 metabolites + APOE4 alleles	0.675 (0.634–0.715)	0.056	0.704	0.375	0.136	0.200	0.076 (0.021–0.126	0.023	0.740	0.375

**Table 8 biomedicines-12-00941-t008:** Prediction of mild cognitive impairment—Dementia with CSF proteins.

Predictors	AUC (95%CI)	Standard Deviation	Accuracy	Precision	Recall	F1	Youden Index	SE	NPV	PPV
3 CSF proteins	0.673 (0.594–0.752)	0.111	0.650	0.600	0.375	0.462	0.163 (−0.004–0.329)	0.074	0.667	0.600
3 CSF protein + APOE4 alleles	0.700 (0.618–0.781)	0.114	0.579	0.333	0.143	0.200	0.193 (0.073–0.313)	0.053	0.625	0.333

**Table 9 biomedicines-12-00941-t009:** Prediction of cognitive unimpaired—Mild cognitive impairment with lipids.

Predictors	AUC (95%CI)	Standard Deviation	Accuracy	Precision	Recall	F1	Youden Index	SE	NPV	PVV
4 lipids	0.655 (0.610–0.701)	0.063	0.797	0.750	0.200	0.316	0.059 (0.012–0.105)	0.021	0.797	0.750
4 lipids + APOE4 alleles	0.680 (0.641–0.719)	0.055	0.597	0.667	0.792	0.724	0.147 (0.0577–0.237)	0.040	0.333	0.667

**Table 10 biomedicines-12-00941-t010:** Prediction of cognitive unimpaired—Mild cognitive impairment with metabolites.

Predictors	AUC (95%CI)	Standard Deviation	Accuracy	Precision	Recall	F1	Youden Index	SE	NPV	PPV
5 metabolites	0.556 (0.516–0.595)	0.056	0.628	0.630	0.983	0.768	0.037 (0.002–0.071)	0.015	0.500	0.630
5 metabolites + APOE4 alleles	0.631 (0.585–0.678)	0.065	0.663	0.663	0.950	0.781	0.079 (0.053–0.104)	0.011	0.667	0.662

**Table 11 biomedicines-12-00941-t011:** Prediction of cognitive unimpaired—Mild cognitive impairment with CSF proteins.

Predictors	AUC (95%CI)	Standard Deviation	Accuracy	Precision	Recall	F1	Youden Index	SE	NPV	PPV
3 CSF proteins	0.760 (0.691–0.828)	0.095	0.700	0.688	0.917	0.786	0.301 (0.196–0.406)	0.046	0.750	0.688
3 CSF protein + APOE4 alleles	0.783 (0.713–0.854)	0.098	0.700	0.688	0.917	0.786	0.389 (0.230–0.548)	0.070	0.7500	0.688

## Data Availability

Data used in the preparation of this article were obtained from the Alzheimer’s Disease Neuroimaging Initiative (ADNI) database (http://adni.loni.usc.edu, assessed 21 December 2023).

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
