# Peer review of "Signatures and Discriminative Abilities of Multi-Omics between States of Cognitive Decline"

_biomedicines, 2024, doi:10.3390/biomedicines12050941_

Round 1
Reviewer 1 Report
Comments and Suggestions for Authors
Dr. Filippos Anagnostakis' paper explores markers that can predict the decline in dementia from various omics of biological samples. The paper is to be commended for the many efforts devoted to it. However, some questions remain.
Different biological samples are measured with different omics analysis methods, but the reasons are unclear. For example, why is the intestinal sample only metabolomic and not proteomic, while CSF is proteomic and not metabolomic? Blood is being genotyped, and lipidomics is being performed for AOPE. Will the CSF be metabolomics only and not proteomics?
Lipidomics in 2.1, metabolomics of enterobacteria in 2.2, and proteomics of CFS in 2.3 only describe the instruments used. Existing papers are cited, but sample processing should be short, and the essence should be described.
As for the samples in 2.1-2.3, it is not clear how reproducibility and quantitation are ensured. Are multiple measurements performed at the sample and measurement levels?
Are sample concentrations diluted to check the linearity between the concentration of each substance and the signal?
Are you checking for ion suppression?
At what level is substance identification performed?
What quality control is used?
How do you compensate for the blurring of data due to the timing of sample collection?
Is the normalized intensity in 2.3 normalized per sample? Or is it some other normalization? If it is a per-sample normalization, only 83 proteins are measured, but can the overall concentration be measured with those proteins?
Which four substances were measured in the ELISA? For what?
The description of the analysis in 2.5 was vague and any reader may not understand what kind of data processing is being done.
You are converting data to Log10, but are you sure that the data are Poisson distributed?
The missing values should have been pre-processed, but there is no description.
You are converting the data to a Z-score, is this per sample? Or is it per observed variable?
If it is per observed variable, there is no description of how the shading is corrected for each sample. In addition, the conversion to Z-score handles both large and small variations in intensity regardless of whether they are large or small, so there is a possibility that data such as noise may be overestimated. Furthermore, suppose there are several outlier-like data with large values. In that case, the mean and standard deviation will be pulled down by these data, so measures should be taken to make the conversion robust to outlier-like samples, or to correct the entire sample before converting to Z-score.
Although 10 CVs are performed in Logistic regression, CVs should be performed 200 times or so with different random number species and evaluation of generalization ability should be performed with median AUC values, 95% of blurring range, etc.
3. As for the results, only the results after the variables were narrowed down in the data analysis are described, and the evaluation of the entire data is not described, making it impossible to determine whether the sampling and measurement were performed at a certain quality. Normally, a heat map or principal component analysis showing the entire data should be conducted to show that there are no unexpected biases other than the designed factors, and then the results should be described after the variables are narrowed down.
The selected substances were also narrowed down after being Z-scored, so it is not clear whether there are relatively large intensity variations or small intensity variations in the overall data.
Line 105. Two consecutive periods.
Author Response
Thank you very much for your valuable insights that render the quality of the paper higher.
We used ADNI, which doesn't provide data for intestinal proteomic or CSF metabolomic features. Unfortunately, APOE is genotyped only in blood in ADNI. I agree, that by having the oppurtunity to enrich the analyses with these suggestions would provide more clues and insights for potential predictions.
I enriched the manuscript with some additional details regarding the sample processing and generally the methodology. Since we used ready-omic data, we provide the necessary links that explain extensively the quality control, ion suppression, sample processing, intensity normalisation, outlier handling
Log10 transformation was conducted to minimise the skewness of the data and make them normally distributed. Instead of Poisson distribution, normalisation of the data was ensured by q-q plots
For the variables used, no missing values were observed. Participants with missing values for any of the variables, were excluded.
The normalisation done from us after log10 transformation was done for every variable (ex lipids). I corrected that in the text
When dealing with large number of variables like in Lipidomics, in order to avoid that outlier data would pull the mean down, we filtered these data with RSD<25% before converting them to z scores as suggested by Hackstadt, et al.
The selection of a CV 10 was done empirically and by observation of other papers. After the cross validation AUC 95% CI, with SD, accuracy, precision, recall F1, Youden index,SE, NPV, and PPV were calculated and presented in the tables
Data before normalisation and log transormation are described extensively in many original research papers and during data generation, so the authors decided that it would be repetitive and not suitable to mention them in this brief report. Heat maps showing all data would be very informative but given the length of variables (ex Lipidomics>700 lipids) its visualisation would be very challenging. ADNI is a public database and any researcher could request the data and reproduce the analyses.
Thank you very much for your precious feedback. I'd be eager to proceed to further additions to improve the paper.
Reviewer 2 Report
Comments and Suggestions for Authors
A very interesting work, well designed, well executed and well written on a large sample trying to look for biomarkers of mild cognitive impairment and Alzheimer's disease with combinations of lipids, gut metabolites, and CSF proteins. The results are somewhat novel. I suggest, as minor change, to add some speculations on the possible relationship of these markers with others such as beta-amyloid and tau proteins in the discussion.
Author Response
Thank you for your comments. I added a sentence with some speculations on the potential of possible associations of multi-omics with CSF neurodegeneration biomarkers such as beta amyloid and tau
Reviewer 3 Report
Comments and Suggestions for Authors
The study conducted by Anagnostakis et al. investigating alterations and diagnostic abilities of baseline blood lipidomic, serum gut microbial metabolomic, and cerebrospinal fluid (CSF) proteomic profiles in individuals with cognitive alterations. The study is innovative and has a good methodological approach.
The main point in question in the study is the importance of evaluating points that can influence the results of the study, such as medications used by participants, associated pathologies and level of physical activity. These points are important to present, as many studies have already found that they can interfere with the degree of cognitive change.
Author Response
Thank you very much for your valuable input.
Indeed, these factors may have an impact on cognitive decline. In the ADNI dataset, these factors contain a large amount of missing values. In the dataset presented, there are no missing values, so the authors thought not to use some imputation technique for that big amount of missing values that these variables present.
Thank you for your understanding
Round 2
Reviewer 1 Report
Comments and Suggestions for Authors
Well revised. In the table, there must be a space before the parentheses.